# Lanthanum Ferrites-Based Exsolved Perovskites as Fuel-Flexible Anode for Solid Oxide Fuel Cells

**DOI:** 10.3390/ma13143231

**Published:** 2020-07-20

**Authors:** Massimiliano Lo Faro, Sabrina Campagna Zignani, Antonino Salvatore Aricò

**Affiliations:** CNR-ITAE, Istituto di Tecnologie Avanzate per l’Energia “Nicola Giordano”, Via Salita S. Lucia sopra Contesse 5, 98126 Messina, Italy; zignani@itae.cnr.it (S.C.Z.); arico@itae.cnr.it (A.S.A.)

**Keywords:** perovskite, electrooxidation, fuel flexibility, renewables, anode

## Abstract

Exsolved perovskites can be obtained from lanthanum ferrites, such as La_0.6_Sr_0.4_Fe_0.8_Co_0.2_O_3_, as result of Ni doping and thermal treatments. Ni can be simply added to the perovskite by an incipient wetness method. Thermal treatments that favor the exsolution process include calcination in air (e.g., 500 °C) and subsequent reduction in diluted H_2_ at 800 °C. These processes allow producing a two-phase material consisting of a Ruddlesden–Popper-type structure and a solid oxide solution e.g., α-Fe_100-y-z_Co_y_Ni_z_O_x_ oxide. The formed electrocatalyst shows sufficient electronic conductivity under reducing environment at the Solid Oxide Fuel Cell (SOFC) anode. Outstanding catalytic properties are observed for the direct oxidation of dry fuels in SOFCs, including H_2_, methane, syngas, methanol, glycerol, and propane. This anode electrocatalyst can be combined with a full density electrolyte based on Gadolinia-doped ceria or with La_0.8_Sr_0.2_Ga_0.8_Mg_0.2_O_3_ (LSGM) or BaCe_0.9_Y_0.1_O_3-δ_ (BYCO) to form a complete perovskite structure-based cell. Moreover, the exsolved perovskite can be used as a coating layer or catalytic pre-layer of a conventional Ni-YSZ anode. Beside the excellent catalytic activity, this material also shows proper durability and tolerance to sulfur poisoning. Research challenges and future directions are discussed. A new approach combining an exsolved perovskite and an NiCu alloy to further enhance the fuel flexibility of the composite catalyst is also considered. In this review, the preparation methods, physicochemical characteristics, and surface properties of exsoluted fine nanoparticles encapsulated on the metal-depleted perovskite, electrochemical properties for the direct oxidation of dry fuels, and related electrooxidation mechanisms are examined and discussed.

## 1. Introduction

Achieving fuel flexibility in fuel cells has been a focus of high-temperature fuel cells since their discovery [1,2]. As Solid Oxide Fuel Cells (SOFCs) operate at temperatures close or higher than 800 °C, their thermodynamics allow the direct use of dry organic fuels or organic fuels in combination with water [3]. However, practical operation requires a pre-treatment of organic fuels to convert hydrocarbon and remove sulfur traces [4]. In parallel, the discovery of new materials and novel cell designs has allowed reducing the operating temperature to 600–800 °C [5,6]. Several studies have been concerned with the development of novel materials [7,8] and in particular novel anodes for advanced SOFCs operating in fuel-flexible mode [9,10,11,12,13].

Fuel flexibility is treated in this review article in relation to a specific category of electrocatalysts.

Although novel SOFCs designs and several new materials have been investigated, Ni in combination with Yttria-stabilized Zirconia (YSZ) still represents today the selected approach for the anode [14]. This is because the Ni-YSZ cermet provides a proper trade-off in terms of electrochemical, thermal, and mechanical properties [15,16,17,18,19,20,21,22]. In particular, the mechanical and thermal properties of Ni-YSZ are rather flexible and adequate to the production chain of anode-supported SOFC cells [23,24,25]. However, the degradation of Ni-YSZ anodes during SOFC operation is frequently reported. This occurs due to the coarsening of Ni particles [16] and to the carbon deposition [20]. Moreover, another drawback is related to the risk of Ni poisoning due to sulfur contaminants and pore blocking associated with the deposition of carbon tar [26,27,28]. Accordingly, large SOFC systems include a fuel pre-reformer and a desulfurizer [15,29,30,31]. An alternative strategy for SOFC systems’ simplification, which has been quite recently adopted, consists of the use of a functional layer coated on the external side of Ni-YSZ anodes [32]. Due to the favorable electronic properties, Ni alloys in combination with doped ceria electrolyte have been widely investigated to replace bare Ni [33,34,35]. The specific advantage relies on a breaking effect of the crystallographic arrangement of Ni atoms, which is responsible for the cracking mechanism of hydrocarbons [36,37]. This effect has been achieved by the inclusion of a different transition metal, not adsorbing carbon, into the reticular scaffold of Ni. The metals selected for this inclusion were somehow inert toward the cracking reaction (e.g., Cu) [33] or capable of modifying the surface electronic density and the lattice distances of the regular Ni–Ni crystallographic network (e.g., Co) [34], or a combination of all these effects (e.g., Fe) [35]. Then, a further optimization has been achieved by adopting different synthesis methods with the aim of improving the homogeneity of the solid solution and consequently reducing the probability of Ni–Ni bonds’ occurrence on the surface [37].

Moreover, specific catalyst functionalization has been made by increasing its oxygen storage capability [38], e.g., by adding oxygen storage additives such as ceria. This is particularly desired to promote the oxidation of a fuel and to extend the “triple phase boundary” [39,40]. These are the sites where the electrochemical reactions take place. Thus, the alloy is generally mixed with doped ceria, which is a well-known oxygen storage material as a consequence of its peculiar redox properties [10]. Although this approach has been revealed as promising for the direct use of dry organic molecules in SOFCs [41,42], to date, there has been limited evidence about the resistance of these Ni alloy–ceria composite catalysts toward sulfur poisoning.

An approach combining both sulfur resistance and organic fuel oxidation capabilities consists of the use of exsolved perovskites at the anode [43]. ABX_3_ is the general chemical formula of these materials. A and B are the sites occupied by cations with different sizes (e.g., La and Sr for the A site and Fe and Co for the B site) and different valence states. X is the site for the oxygen anions [44]. These materials may be tuned in both A and B sites by using different cations and doping in order to achieve the desired amount of holes and vacancies that are required for high ionic and electronic conductivity [45,46,47]. The idea of using perovskites at the anode is more recent. It raised from the large amount of evidence about their activity toward the cleavage of C–H, C–C, and C=C bonds [46,48,49] with a relatively good stability in the presence of sulfur-based products [50,51]. These systems are also characterized by mixed electronic and ionic conductivity [52], even if the electronic conductivity is not comparable to that of a metal also at the high operating temperatures of an SOFC. On the other hand, the perovskites may present the drawbacks of a limited structural stability under the reducing environment of an SOFC anode [53,54]. To address this issue, the modification of perovskites ex ante with the aim of functionalizing their surface and tailoring the activity for the anodic reactions, while stabilizing the structure and improving the electronic conductivity, has been carried out [43,55]. A further step forward in tailoring these perovskite materials for the use as an anode in SOFCs is relying on the addition of ceria to increase the oxygen storage capacity and the electronic/ionic percolation.

The authors of this mini-review have intensively worked on modified and exsolved perovskite for application as an anode in SOFCs for several years demonstrating an effective oxidation of various organic fuels directly fed to the cell [56,57,58,59,60,61,62,63]. The proper modification of commercial type ferrite-based perovskites, which are commonly used as an SOFC cathode, with small amounts of Ni has been demonstrated. The process has been enhanced by tailoring the thermal treatments to consolidate the modified perovskite structure. In this mini-review, we have summarized the properties of these novel anode materials, their performance in SOFCs, and the specific reaction mechanisms investigated through a combination of electrochemical experiments and the chromatographic analyses of effluents.

## 2. Surface Exsolution and Physicochemical Studies

The exsolution of fine particles from the perovskite surface is an approach recently used for the functionalization of perovskite materials for anodic applications [64,65,66]. Two main approaches have been developed: (1) synthesis of a non-stoichiometric perovskite host [67] and (2) surface impregnation of a raw perovskite with a foreign transition element [68,69]. The final modification is thermal driven by a process under reducing condition at high temperature in order to achieve a proper thermodynamic stability for the fine exsolved nanoparticles embedded in the perovskite substrate. The latter remains deficient of cations in A and B sites. Both approaches have been addressed to promote a similar distribution of fine exsolved particles on the surface. It was reported that the first method should be preferred in case of high risks for their coarsening [70].

Many exsolved perovskites have been synthesized using these approaches for various applications and with the aim of functionalizing the surface of these materials. One of the most used applications has regarded their use as a novel anode for SOFCs. The subject of this mini-review is specifically addressing the modification of ferrite-based perovskites commonly used as cathodes in commercial-type SOFCs operating in the temperature range between 700 and 800 °C. This type of perovskite generally has the formula La_0.6_Sr_0.4_Fe_0.8_Co_0.2_O_3_ (LSFC). The raw LSFC that we have largely used in our previous works had a surface of 5.20 m^2^ g^−1^ and was purchased from Praxair. By using a wet impregnation method, 20 cm^3^ of an aqueous solution containing 3 wt % of Ni as Ni nitrate (Sigma Aldrich) has been deposited drop-by-drop on the perovskite, and this was maintained under stirring condition at 80 °C. After drying at 150 °C for 8 h, the powder has been treated at 500 °C in static air for 2 h and then reduced with 5% of H_2_ in N_2_ at 800 °C for 2 h and then re-calcined at 500 °C in static air for 2 h to stabilize the nanoparticles on the surface. The amount of Ni added to the perovskite was evaluated on the basis of a compromise between the need of avoiding large occurrence of metallic nickel on the surface with the consequent risk of carbon deposition during the cracking of organic fuels and to favor just a partial exsolution of Fe and Co from the perovskite. The subsequent step consisted in the grinding for 12 h of this powder together with gadolinia-doped ceria (Gd_0.1_Ce_0.9_O_2_-GDC, Praxair) with a surface area of 38.92 m^2^ g^−1^. The doping of perovskite with Ni and its subsequent thermal treatments caused a modification of the initial perovskite phase originated from the depletion of Co and Fe with the consequent formation of a new phase named as n = 1 Ruddlesden–Popper (A_n+1_B_n_O_3n+1_) structure [71,72]. Figure 1 shows the XRD spectra of the raw and modified perovskites. Fe and Co are located in the blue octahedrals, and the green spheres are related to the sites of La and Sr, whereas the red spheres are the sites for oxygen ions. As a consequence of perovskite distortion, the ratio La/Sr (0.6/0.4) remained the same during the perovskite modification, whereas the Fe/Co ratio was altered as a consequence of the Co and Fe depletion from the bulk and their migration to the perovskite surface (the Fe/Co ratio was originally 4 in the raw perovskite). The overall amount of Ruddlesden–Popper n = 1 phase was approximately 60 wt % as determined through a least-square fitting profile quantitative analysis procedure. As evidenced by previous High Resolution—Transmission Electronic Microscopy (HR-TEM) and Energy-Dispersive X-ray (EDX) analyses [62], the exsolution process involves the occurrence of a secondary phase related to fine exsolved particles encapsulated on the surface. These are composed of a tri-metallic alloy (e.g., Ni-Fe-Co, about 25 nm) in the core and a shell of mixed oxides (e.g., α-Fe_100-y-z_Co_y_Ni_z_O_x_, about 2 nm). Smith et al. [73,74] have reported that this mixed oxide has shown reversible electrochemical behavior starting from low temperatures. Accordingly, it has been also suggested as a substitution of noble metals for the water electrolysis in zero-gap cells [73,74].

The redox properties of these modified perovskites and their exsoluted fine particles have been the object of specific studies [57]. Figure 2 shows the temperature-programmed reduction (TPR) profile of Ni-LSFC/CGO (i.e., Ce_0.8_Gd_0.2_O_2−d_), the HR-TEM image of a fine particle encapsulated in the surface of the catalyst and its EDX analysis. As discussed, Co and Fe were alloyed with Ni and migrated to the surface of perovskite as proved by the EDX analysis carried out during high-angle annular dark-field (HAADF) imaging, whereas an oxide shell was confirmed by the combination of EDX and HR-TEM analysis. This peculiar structure imparts to the material outstanding catalytic properties [73,74]. The TPR profile showed three signals in the temperature range of 300–650 °C related to the chemisorption of H_2_ on Co (α_1_ [75]), Fe (α_2_ [76]), and Ni (α_3_ [77]). This indicates that the oxide surface of these nanoparticles turns into a trimetallic system under operation in reducing conditions at high temperature. The TPR profile showed also a broad peak around 650 °C, which is due to the reduction of Ce (Ce^4+^ → Ce^3+^, α_4_ [78,79]). Based on the XPS studies [62], the most probable oxidation states of metals for the fine embedded particles was 3^+^ for both the cobalt and the iron and 2^+^ for the nickel. By integrating the three TPR peaks observed in the temperature range between 300 and 650 °C, the following composition α-Fe_23_Co_15_Ni_12_O_x_ is derived, suggesting an excess of iron in line with the evidence of the EDX profile.

Generally, the surface area of these exsolved perovskite materials is 5.51 m^2^ g^−1^ as evaluated by Brunauer–Emmett–Teller (BET) analysis with mesoporous morphology [58]. Thus, the formation of fine embedded particles does not promote a significant increase of the overall surface roughness for the modified perovskite catalyst; instead, the initial surface area is quite similar (e.g., 5.2 m^2^ g^−1^).

## 3. Catalytic Studies

As this catalyst was suggested for the electrochemical conversion of organic fuels in SOFCs, preliminary catalytic tests have been carried out to collect information concerning its reliability toward the most common reactions occurring in the anode compartment of an SOFC during its operation. It has been largely reported that the most frequent reactions occurring at the anode compartment fed with organic fuels can be simulated ex situ by the autothermal reaction (ATR) [80,81]. In particular, water besides CO_2_ is formed during the oxidation of organic fuels and oxygen is provided at the electrolyte–anode interface. Nevertheless, some specific differences exist between the ATR reaction and the electrochemical reaction. This essentially concerns the fact that molecular oxygen is supplied during the ATR, whereas ionic oxygen is involved in the redox process. Thus, the net reactions of an electrochemical process and ATR reaction are similar but with the difference that the electrochemical reaction involves ionic oxygen and the release of electrons into the external circuit. It has been discussed in the literature that an SOFC may be fed with oxygen or water in combination with organics in order to minimize the risks associated with the carbon fibers’ deposition (i.e., Partial Oxidation Reaction (POX) [82,83], Steam Reforming (SR) [82,84], and a combination of dry reforming and ATR named as tri-reforming [85,86], respectively). Although a right balance of the stoichiometry may effectively minimize this risk, other aspects associated to the dilution of fuel, lower efficiency, and re-oxidation of the anode have been suggested to avoid the in situ supply of molecular oxygen [4,84,87,88,89,90].

Generally, the risk of damaging an SOFC increases with the increasing of the carbon atoms in the fuel molecule. This can be mitigated by varying the steam-to-carbon and oxygen-to-carbon ratios. Therefore, the catalytic behavior of the exsolved perovskite materials has been preliminarily investigated under POX, SR, and ATR of methane [59], methanol [58], propane [59], and glycerol [57]. According to such previous studies, their reactivity toward oxidation follows this trend: methanol > ethanol > propane > glycerol > methane (Table 1). In particular, it has been shown [59] that methane did not reach much at 800 °C, whereas other organics showed a significant production of syngas, with an H_2_/CO ratio that varied according to the H/C ratio in the fuel. The same work also reported that the presence of unreacted fragments (i.e., CH_4_) was dependent on the molecular weight of the fuel. In addition, contrary to what was reported in the literature for the propensity of LSFC to catalyze the oxidative coupling of methane [54], no C2 compound has been detected in the outlet stream.

Specific works in the literature have addressed the stability of the perovskite in the presence of sulfur contaminants. Exsolved perovskite was investigated for the ATR of propane in the presence of H_2_S up to 80 ppm. These studies [60] showed that the increase of sulfur in the feed caused a depletion of H_2_ in the produced syngas and a decrease of propane conversion, since H_2_S has blocked the active sites of the electrocatalyst. Nevertheless, the overall negative effects caused by the H_2_S appeared to be reversible, since the spent catalysts did not show any significant changes in their structure as revealed by XRD as well as limited amount of carbon and sulfur contents was detected through the elemental analysis (CHNS-O). In principle, the catalysts could be regenerated by proper activation.

## 4. Electrochemical Studies

According to the ex situ catalytic studies, the modified perovskites appear to be promising catalysts for application in a fuel-flexible SOFC. This hypothesis is supported by the different electrochemical behavior of LSFC and Ni-LSFC toward the electrochemical oxidation of dry propane [56]. This material has been investigated extensively in two main cell configurations consisting of a single fuel electrode supported on three different types of supporting electrolytes (e.g., CGO [56], La_0.8_Sr_0.2_Ga_0.8_Mg_0.2_O_3_ (LSGM) [61], and BaCe_0.9_Y_0.1_O_3-δ_ (BYCO)) and as a functional layer (pre-layer) for the anode of a commercial type cell (ASC-400B, ELCOGEN [62,63]). In the latter case, the SOFC consists of a dual-anode configuration where the modified perovskite acts as a catalytic pre-layer for the conversion of the organic molecules before these can reach the supporting Ni-YSZ anode.

The electrochemical experiments for the modified perovskite-based SOFC, carried out in H_2_, are reported in Figure 3a,b. The polarization curves (Figure 3a) reveal that the highest performance has been achieved by using the exsolved perovskite as a protective or pre-layer layer in an anode-supported cell configuration. The latter was based on the ASC-400B-Elcogen commercial cell simply modified by coating the exsolved perovskite on the outer anode surface. The highest performance achieved with the modified Elcogen cell is essentially due to a combination of lower ohmic constrain and higher open circuit voltage (OCV). Limited ohmic losses were also observed for the test carried out with a CGO electrolyte-supported cell (series resistance, Rs = 0.214 ohm cm^2^ @ 0.7 V for a dense electrolyte of a thickness of about 250 μm). However, in this case, the maximum achievable performance was limited by the low OCV. The latter was due to the mixed conductivity phenomenon of CGO, which corresponds to an internal current drag [91]. A different behavior was observed for the LSGM supporting cell. In this case, the achieved OCV was even higher than the thermodynamic value for water splitting at 800 °C. This aspect has been attributed to an additional oxygen pump effect that adds about 100 mV to the theoretical Nernst potential for the main reaction [92]. Despite this advantageous OCV value, the achieved performance was affected by a slight higher ohmic resistance (Rs = 0.32 ohm cm^2^ @ 0.7 V achieved with a dense electrolyte of a thickness of about 300 μm). Such lower performances for the electrolyte-supported SOFCs can be mitigated by reducing the electrolyte thickness, since the oxygen ion conductivities of CGO and LSGM are similar and better than the YSZ used in the Elcogen-modified cell [8,61,93]. However, this should not occur at the expense of the mechanical robustness. It is pointed out that the thickness of the overall anode-supported Elecogen cell is much larger than the electrolyte-supported cells. However, the excellent electronic conductivity of the anode support avoids the occurrence of relevant ohmic losses.

Another positive aspect associated with the use of LSGM consists of the possibility of achieving a proper mechanical and chemical compatibility with the exsolved perovskite being characterized by a similar structure and allowing the formation of a complete perovskite structure-based cell [61]. The polarization curve obtained with a protonic BYCO electrolyte indicates a proper OCV denoting an optimal sealing of cell and the presence of additional oxygen conductivity for this electrolyte. In this case, no electronic drag associated to the redox behavior of cerium species occurs, since Ce^4+^ is stabilized in the orthorhombic phase of the perovskite [94,95]. However, the performance achieved for the BYCO-based cell was strongly affected by large ohmic losses (1.44 ohm cm^2^ @ 0.7 V for a dense electrolyte with a thickness of about 300 μm). This was in part due to the carbonation reaction of Ba generally occurring during the severe thermal treatment needed for its densification (above 1300 °C), which promotes the formation of an insulating phase [96,97]. Figure 3b shows the first derivate of the I–V curves of Figure 3a, highlighting the change of area surface resistance versus the current density. It is noteworthy that the typical activation control at low current density is almost absent for all experiments except for the case of the experiment carried out with the LSGM electrolyte. In this case, a fast decrease of resistance occurred during the first 300 mA cm^−2^, and this is due to the extra potential (about 100 mV) caused by the oxygen pump effect discussed above. By observing the Area Specific Resistance (ASR) curve of the experiment carried out by the BYCO electrolyte, a stable and high resistance constraint was observed, confirming the evidences reported above. According to these experiments, the exsolved perovskite may be properly used as an anode for SOFCs fed with H_2_ provided it is combined with a proper electrolyte material. In the case of the Elcogen-modified cell, a peak power density of 1.4 W cm^−2^ is achieved, whereas at 0.8 V, the current density largely exceeds 1 W cm^−2^.

However, the modified perovskite catalysts have been mainly developed for the electro-oxidation of organic compounds with the aim to develop a fuel-flexible SOFC cell. Experiments reported in the literature have concerned the use of dry fuels (e.g., methane, syngas, methanol, ethanol, propane, and glycerol). Polarization curves for the modified perovskite-based cells directly fed with various organic fuels are shown in Figure 4a,b. One of the first reports addressing fuel flexibility was published in 2012 [59]. Electrochemical tests dealing with the direct oxidation of various dry organic fuels were carried out for about 130 h without any relevant evidence of carbon formation. Figure 4a shows the cell performance behavior for the direct oxidation of several organic fuels using a thick (250 µm) CGO–electrolyte-supported cell containing a modified perovskite anode. The best performance is achieved for syngas, propane, and methanol, whereas low performance is observed for methane. However, despite the large thickness of the supporting electrolyte, the influence of the internal redox properties of the ceria-based electrolyte (Ce^4+^ → Ce^3+^), in the presence of a reducing environment at the anode, dominates the polarization behavior. As expected, the most effective electronic state of Ce depends on the reducing environment. As a consequence, on the cathode, the oxidation state is Ce^4+^, whereas on the anode side, this is Ce^3+^. The migration of oxygen from the cathode to the anode has the effect of moving the equilibrium between these two oxidized states of Ce. This mechanism corresponds to an internal chemical “short circuit” for the CGO electrolyte.

As a consequence, the observed OCV values are below 1 V for all experiments shown in Figure 4a denoting an internal electronic drag due to the CGO redox properties [91], as discussed above. Moreover, the performances are also negatively affected by the thickness of the CGO electrolyte.

The low performance achieved in the presence of dry methane is related to a poor reactivity of this molecule over the modified perovskite surface as corroborated by the low OCV. In this case, the polarization curve is strongly affected by both activation and ohmic constrains. Differently from other organic fuels, direct methane oxidation appears quite difficult at the perovskite surface as confirmed by ex situ catalytic tests carried under authothermal reforming (ATR) mode for methane [59].

Despite these limitations, the overall stability shown by this cell configuration during endurance tests with dry organic fuels was promising and prompted further investigation with more appropriate cell designs. Figure 4b summarizes some polarization curves achieved with different dry organic fuels and cell designs all based on modified perovskite anodes. One can observe that the cell designs can dramatically affect the performance. Ceria-based electrolytes show poor performance as a consequence of the significant electronic drag and low OCV, while the anode-supported cell with thin YSZ electrolyte shows the best performance even if the OCV is slightly lower than that of the equivalent cells fed with hydrogen, supporting the notion that LSGM electrolyte-based cells show excellent OCV but strong ohmic losses. Thus, the most appropriate approach is to coat Ni-YSZ anode-supported cells with a modified or exsolved perovskite pre-layer (protecting layer or conducive catalytic layer). This can enhance fuel flexibility without relevant modifications of the SOFC technology.

For most of the cell configurations, maximum power densities are achieved at a potential of around 0.5 V. This voltage is generally considered as the lower potential limit for SOFC operation. At lower voltages, the overall efficiency is dramatically affected by the reduced voltage efficiency. This indicates that further efforts should be addressed to improve the voltage efficiency of dry organic fuel-fed SOFC cells by further amelioration of the modified perovskite properties. Moreover, it is necessary to acquire more insights into the poor reaction kinetics of methane oxidation at the perovskite catalyst surface to improve the anode characteristics.

Several experiments carried out with organic fuels are summarized in Table 2. In one case, the exsolved perovskite has been studied for 780 h in the presence of a large excess of dry propane observing a decay of 1.1 × 10^−4^ A h^−1^. In principle, a large excess of fuel suppresses the partial pressure of water and CO_2_ formed during the electrochemical reaction occurring at the anode. As a consequence, the wet and dry reforming reactions are limited, and the reactions causing the cracking of the organic fuel become largely probable. The open circuit voltage condition appears to be the most stressful situation for the anode when this is fed with organic fuels. Therefore, a demonstration of SOFC stability for a time exceeding 100 h under this condition is considered to be a very promising result.

Prolonged operation has not damaged the SOFC cell as well as the anode, and there was no significant occurrence of carbon deposits on the surface [56]. Significant advances have regarded the use of this electrocatalyst as a coating layer of a commercial-type cell fed directly with dry organic fuels such as ethanol and glycerol. The effective oxidation of these fuels has confirmed the perspective that this electrocatalyst may have potential for the simplification of this technology, in particular of the balance of the plant by reducing the fuel-processing steps. Since the approach, consisting of the use of a proper coating layer of modified perovskite on the anode, does not imply the modification of the SOFC production chain while making these devices fuel-flexible, a proper diffusion of this approach is envisaged in the next years. Regarding the role of CGO mixed to the perovskite electrocatalyst, a previous paper [56] has elucidated the promoting effect of this mixed ionic electronically conducting material for propane oxidation.

In general, various results have demonstrated a strong increase of electrocatalytic activity in the presence of the exsolved perovskite. However, the activity is essentially related to the molecule structure and reactivity with the best results obtained with ethanol and propane (Table 2). In particular, in the presence of methane in a test carried out with a CGO-supported cell, a very low performance has been obtained [59]. The possible reasons for this behavior have been discussed in light of the electrochemical and chromatographic analysis results.

## 5. Reaction Mechanism

One important aspect of the previous studies carried out on modified perovskite electrocatalysts has regarded the reaction mechanisms involving the direct oxidation of dry organic fuels [62,63]. Figure 5 depicts the mechanisms suggested for the electrochemical conversion of ethanol. The oxidation of dry ethanol appears to proceed through multiple and consequent reactions. These involve dehydrogenation (Equations (1)–(3)) and decomposition (Equation (4)) steps with a consequent electrochemical oxidation of H_2_ (Equation (5)) and CO (Equation (6)) to form water and CO_2_, respectively.

The complete process is described below:(1)$${\mathrm{CH}}_{3}{\mathrm{CH}}_{2}\mathrm{OH}\;\overset{\mathrm{Ni}-\mathrm{Perovskite}/\mathrm{CGO}}{\to }-{\mathrm{CH}}_{2_{\mathrm{ads}}}+\mathrm{CO}+2{\;H}_{2}$$
(2)$${\mathrm{CH}}_{3}{\mathrm{CH}}_{2}\mathrm{OH}\;\overset{\mathrm{Ni}-\mathrm{Perovskite}/\mathrm{CGO}}{\to }{\mathrm{CH}}_{3}\mathrm{CHO}+H_{2}$$
(3)$${\mathrm{CH}}_{3}\mathrm{CHO}\;\overset{\mathrm{Ni}-\mathrm{Perovskite}/\mathrm{CGO}}{\to }{\mathrm{CH}}_{4}+\mathrm{CO}$$
(4)$$H_{2}O+-{\mathrm{CH}}_{2_{\mathrm{ads}}}\;\overset{\mathrm{Ni}-\mathrm{Perovskite}/\mathrm{CGO}}{\to }2{\;H}_{2}+\;\mathrm{CO}$$
(5)$$2{\;H}_{2}+2{\;O}^{2-}\overset{\mathrm{Ni}-\mathrm{YSZ}}{\to }2{\;H}_{2}O+4{\;e}^{-}$$
(6)$$2\;\mathrm{CO}+2{\;O}^{2-}\overset{\mathrm{Ni}-\mathrm{YSZ}}{\to }2{\;\mathrm{CO}}_{2}+4{\;e}^{-}.$$

The latter processes occur at the triple phase boundaries of the Ni-YSZ anode. This mechanism has been named a “shuttle mechanism” to indicate the migration of H_2_ and CO, which are produced at the pre-catalytic layer according to an internal reforming step, to the interface and their return to the pre-catalytic layer as H_2_O and CO_2_ favoring the fuel processing. The side reforming reactions involving water and CO_2_ also play a role even if to a lower extent in the presence of an excess of fuel. Such a shuttle mechanism has been suggested based on a combination of chromatographic and HPLC (high-performance liquid chromatography) studies carried out on the gaseous and liquid products for a cell operating with dry ethanol. These analyses have revealed the presence of acetaldehyde and acetic acid in addition to CH_4_, CO_2_, H_2_O, and CO. These evidences have proved that one of the possible rate-determining steps for this oxide catalyst regards the dehydrogenation. Therefore, it is largely probable that the cell fed with methane has shown lower performance, since its dehydrogenation is more difficult than that of other molecules. In addition, the oxidative coupling sometimes reported as a possible reaction catalyzed by perovskites [98] including the LSFC [54] may be reasonably neglected, since no carbon-based molecules with higher molecular weight than methane have been observed in the chromatographic analyses.

## 6. Key Insights on Challenges and Perspectives

The ferrite-based perovskite used as a substrate for a pre-catalytic layer is currently a benchmark for the oxygen electrode in Solid Oxide Fuel Cells (SOFCs) and Solid Oxide Electrolysis Cells (SOECs) operating at intermediate temperature. Specific strengths are the lower cost and the lower constraints compared to the manganite-based perovskites [99,100,101]. Therefore, its use as a fuel electrode or as a promoter for an in situ fuel processor is highly desired [102]. However, this material is not sufficiently stable in a reducing environment [54,103]. It may become chemically stable upon reduction at high temperature and utilization at intermediate temperature. Moreover, as for the most perovskite, the rates for the electroxidation of H_2_ and organic fuels are low. A possible solution to this issue is the decoration with metallic-based particles as discussed above. Based on the experiments reported in the literature, the modified Ni-LSFC is a promising electrocatalyst for organic fuels that may be easily dehydrogenated at the very early stage of their oxidation. On the other hands, Ni-LSFC has shown strong limitations for the conversion of CH_4_. Several papers have discussed the use of Ni alloys as a possible electrocatalyst for the oxidation of light organic fuels such as methane [104,105,106]. This allows addressing the low reactivity of exsolved perovskites toward methane, making the pre-catalytic layer more suitable for a fuel-flexible SOFC. We have recently started the investigation of a novel exsolved LSFC by adding 30 wt % of a NiCu alloy (1:1 at.). This electrocatalyst was prepared with the same procedure as that mentioned above. A conventional anode-supported cell (Elcogen) modified with such composite catalyst has shown very promising performance and durability. The cell was fed with dry biogas (30 vol.% CO_2_ and 70 vol.% CH_4_), and it showed > 0.2 W cm^−2^ with biogas and a low decay during a lifetime test carried out at 650 °C for about 100 h.

Figure 6a shows the polarization curves achieved with this novel composite electrocatalyst. The cell has shown an OCV close to 1 V in the presence of biogas. The achieved cell performance approached 200 mW cm^−2^ at 0.66 V with biogas. The curves seem to be affected only by ohmic constraints due to the intermediate temperature operation (i.e., 650 °C). Figure 6b shows a durability time test at a current density of 150 mA cm^−2^ with a high fuel flow rate (i.e., 5 cc min^−1^ cm^2^). The cell appears very stable under dry biogas feed and no carbon deposition is envisaged under these experimental conditions.

This novel approach should be evaluated in relation to the limited performance achieved for the Ni-LSFC fed with dry methane and reported in Figure 4a. As we discussed above, Ni-LSFC showed a low reactivity toward the oxidation of methane, and this is ascribed to the rate-determining step due to the dehydrogenation of this fuel. The present composite catalyst (NiCu-LSFC) registered an OCV close to 1 V, which proves a high reactivity toward CH_4_. This is in line with the reactivity we have already demonstrated for various Ni-based alloys in several recent papers [11,12,33,34,37,107]. However, some mass transfer limitations at a cell voltage of about 0.6 V have been observed. This effect has a negative impact on the “shuttle mechanism”, causing such voltage decay at sustained currents in the polarization curves.

## 7. Conclusions

The analysis presented in this mini-review deals with a number of studies carried out over more than a decade on a novel modified perovskite used as an anode electrocatalyst for application in fuel-flexible SOFCs. The exsolved perovskite shows the presence of fine embedded particles made of transition elements such as Fe, Ni, and Co over a depleted perovskite substrate. This structure is characterized by outstanding catalytic activity and proper electronic conductivity under reducing conditions. Moreover, a limited coarsening effect is generally observed for the embedded nanoparticles even upon prolonged operation. Catalytic and electrochemical studies have proved an effective capability of this material to convert organic fuels and good chemical stability under reducing conditions. These evidences make this electrocatalyst a promising anode for electrolyte-supported cells or as a coating layer for anode-supported cells. The exsolved perovskite catalyst can also be modified by the addition of an Ni alloy to form a composite layer. A specific “shuttle mechanism” is envisaged for the catalyst operation in an SOFC. This includes iterative adsorption and desorption steps involving the organic fuel and the reaction intermediates, the reaction at the interface of the formed hydrogen and carbon monoxide, with the final formation of water and CO_2_. This approach appears very promising for multifuel-fed SOFCs, and it is expected to attract large interest in the coming years.

## Figures and Tables

**Figure 1 materials-13-03231-f001:**
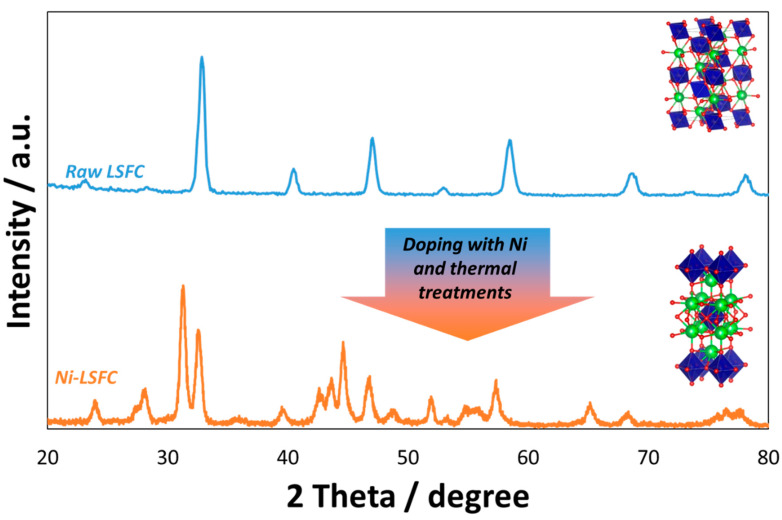
Structural analysis of raw (top) and modified (bottom) perovskites. The figure also reports the related unit cells. As shown, the addition of Ni and subsequent thermal treatments (calcination at 500 °C and reduction at 800 °C) caused the intercalation of one rocksalt-type phase into the perovskite (n = 1 Ruddlesden–Popper phase).

**Figure 2 materials-13-03231-f002:**
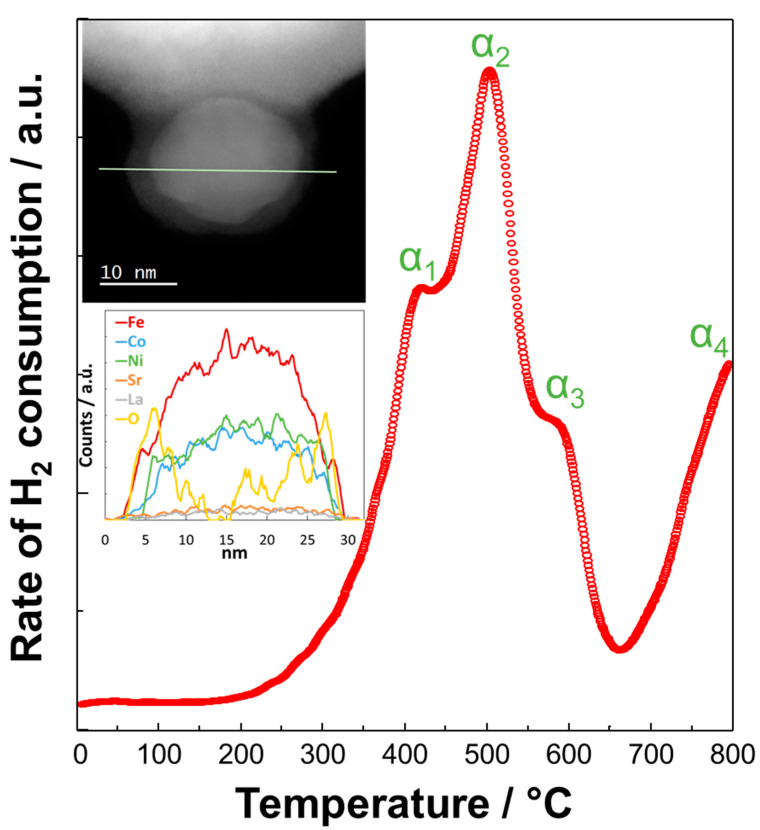
Temperature-programmed reduction (TPR) profile of Ni-LSFC/CGO, TEM image, and the Energy-Dispersive X-ray (EDX) measure carried out along the profile of an embedded nanoparticle on the surface of a modified perovskite. CGO: LSFC: La_0.6_Sr_0.4_Fe_0.8_Co_0.2_O_3_.

**Figure 3 materials-13-03231-f003:**
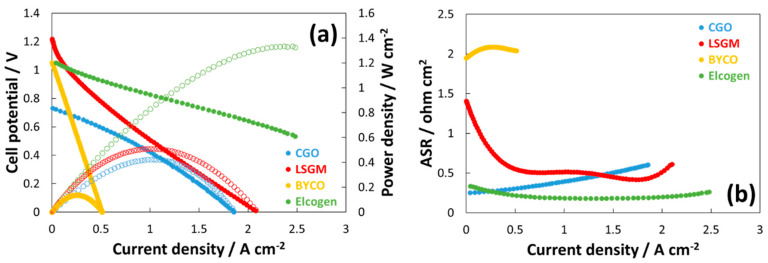
Polarization and ASR curves at 800 °C for the modified perovskite-based cells fed with H_2_ using three supporting electrolyte based Solid Oxide Fuel Cells (SOFCs) (e.g., CGO, LSGM, and BYCO) and a commercial Elcogen cell coated at the anode with an exsolved perovskite pre-layer. (**a**) Cell potential and power densities vs. current density; (**b**) area specific resistance vs. current density. (Standard deviation was < 7 mV for each data point of the overall polarization dataset).

**Figure 4 materials-13-03231-f004:**
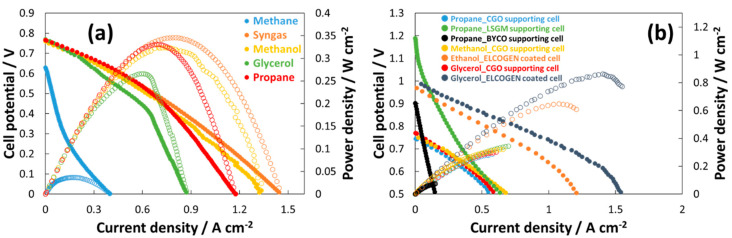
Polarization curves for the modified perovskite-based cells operating in multifuel mode in the presence of a thick supported CGO electrolyte (**a**) adapted from [59] and with different organic fuels and cell designs (**b**) adapted from [56,57,58,61,62,63]. (Standard deviation was < 5 mV for each data point of the overall polarization dataset).

**Figure 5 materials-13-03231-f005:**
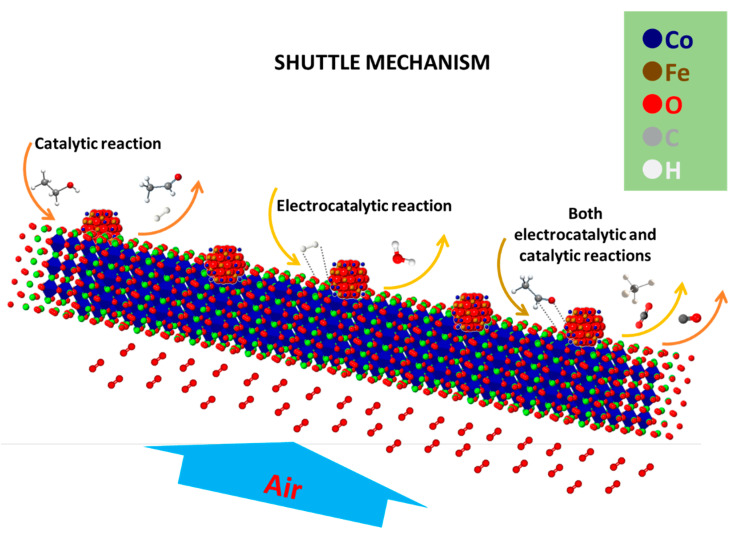
Exemplification of the shuttle mechanism suggested for dry ethanol fed at a modified perovskite-based SOFC cell. The support represents the exsolved perovskite, whereas the fine embedded particles are ascribed to the α-Fe100-y-zCoyNizOx oxide. Air is fed to the cathode, and oxygen ions migrate to the anode through the electrolyte.

**Figure 6 materials-13-03231-f006:**
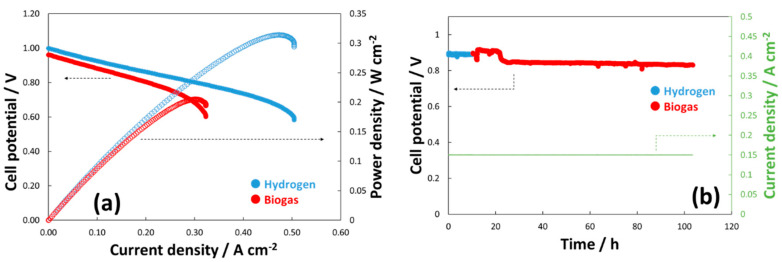
Polarization curves (**a**) and lifetime test (**b**) carried out with a novel NiCu-exsolved La_0.6_Sr_0.4_Fe_0.8_Co_0.2_O_3_ (LSFC) perovskite composite catalyst. These tests have been carried out at 650 °C with NiCu-LSFC coated on an Elcogen cell fed with H_2_ and biogas (30 vol.% of CO_2_). (Standard deviation was < 5 mV for each data point of the overall polarization dataset).

**Table 1 materials-13-03231-t001:** Survey of the average product formation achieved in different reaction processes at 800 °C. Data from [57,58,59,60].

		H_2_ %	CO %	CO_2_ %	CH_4_ %	C_2_H_4_ %	C_3_H_8_ %	Others %	Selectivity to Syngas %
SR-800 °C	Glycerol (S/C = 0.2) [57]	29.45	28.39	32.90	7.50	1.76	-	-	-
SR-800 °C	Glycerol (S/C = 2) [57]	80.79	5.60	2.36	5.82	5.43	-	-	-
ATR-800 °C	Methane (S/C = 2.5; O/C = 0.5) [59]	5.95	2.19	1.62	85.68	-	-	-	-
ATR-800 °C	Methanol (S/C = 2.5; O/C = 0.5) [58,59]	67.44	13.73	17.71	0.56	-	-	-	87.42
ATR-800 °C	Propane (S/C = 2.5; O/C = 0.5) [59,60]	66.59	17.40	4.94	6.98	2.35	0.97	0.77	-
ATR-800 °C	Glycerol (S/C = 2.5; O/C = 0.5) [59]	31.10	29.62	29.01	9.56	-	-	-	-
SR-800 °C	Methanol (S/C = 2.5) [58]	-	-	-	-	-	-	-	81.70
POX-800 °C	Methanol (O/C = 0.5) [58]	-	-	-	-	-	-	-	92.02
SR-800 °C	Propane (S/C = 2.5) [60]	64.59	15.04	9.34	6.75	1.99	1.48	0.81	-
POX-800 °C	Propane (O/C = 0.5) [60]	43.73	29.20	0.68	13.03	10.06	1.68	1.62	-
ATR-800 °C	Propane (S/C = 2.5; O/C = 0.5) + 20 ppm H_2_S [60]	22.38	8.73	18.76	15.48	28.57	6.18	-	-
ATR-800 °C	Propane (S/C = 2.5; O/C = 0.5) + 40 ppm H_2_S [60]	23.97	10.29	18.07	16.28	25.02	6.37	-	-
ATR-800 °C	Propane (S/C = 2.5; O/C = 0.5) + 60 ppm H_2_S [60]	14.41	10.62	19.75	15.91	26.38	10.14	2.79	-
ATR-800 °C	Propane (S/C = 2.5; O/C = 0.5) + 80 ppm H_2_S [60]	13.25	8.76	19.60	16.55	29.06	12.74	0.04	-

**Table 2 materials-13-03231-t002:** Resume of the most important results of the electrochemical tests carried out at 800 °C with different organic fuels. Data from [56,57,58,59,61,62,63].

	Type of Cell/Electrolyte	Maximum Power Density/mW cm^−2^	Series Resistance /Ω cm^−2^	Total Resistance /Ω cm^−2^	Maximum Durability Demonstrated/h	Average Decay during the Life Time Test/A h^−1^
Methane [59]	CGO (250 μm)	37 @ 0.22 V	0.51 @ 0.5 V	2.78 @ 0.5 V	15	0
Syngas [58]	CGO (250 μm)	346 @ 0.44 V	0.24 @ 0.765 V	0.29 @ 0.765 V	17 [59]	0
Methanol [58]	CGO (250 μm)	358 @ 0.47 V	0.26 @ 0.75 V	0.33 @ 0.75 V	18 [59]	4 10^−3^
Ethanol [62]	Elcogen	648 @ 0.60 V	0.18 @ 0.7 V	0.46 @ 0.7 V	400	1.5 10^−4^
Propane [56]	CGO (250 μm)	288 @ 0.51 V	0.25 @ 0.5 V	0.33 @ 0.5 V	780	1.1 10^−4^
Propane [61]	LSGM (300 μm)	328 @ 0.43 V	0.32 @ 0.7 V	0.92 @ 0.7 V	15	5 10^−4^
Glycerol [57]	CGO (250 μm)	320 @ 0.46 V	0.30 @ 0.5 V	0.66 @ 0.5 V	19 [59]	0
Glycerol [63]	Elcogen	864 @ 0.62 V	0.12 @ 0.7 V	0.25 @ 0.7 V	157	1 10^−3^

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
