# Peer review of "Lanthanum Ferrites-Based Exsolved Perovskites as Fuel-Flexible Anode for Solid Oxide Fuel Cells"

_materials, 2020, doi:10.3390/ma13143231_

Round 1
Reviewer 1 Report
This article is a mini-review about a certain type of perovskites as fuel cell materials.
It is excellent to nicely mention currently important materials from a keen view point. So it should be accepted almost as it is.
However, some confusion (figure(s) and Fig(s).) was observed in the manuscript. Please check at least at the proof stage.
That's all.
Author Response
R. This article is a mini-review about a certain type of perovskites as fuel cell materials.
It is excellent to nicely mention currently important materials from a keen view point. So it should be accepted almost as it is. However, some confusion (figure(s) and Fig(s).) was observed in the manuscript. Please check at least at the proof stage.
A. We thank the reviewer for the useful comments. Concerning the observations raised on the manuscript, we have improved the text and replaced “Fig.” with “Figure”, accordingly.
Reviewer 2 Report
A mini review on lanthanum ferrites-based exsolved perovskites as fuel-flexible anode for Solid Oxide Fuel Cells is very interesting paper.
Some improvements are required:
Line 15 "of the nanoparticles exsoluted on the substrate surface can be further modulated by a post treatment". Which nanoparticles?; Which size?
Line 18: at the SOFC anode (please to write "solid oxide fuel cell (SOFC)
Line 26: Which core-shell nanoparticles?
Line 86: perovskites commonly used as a SOFC cathode with small amounts of Ni has been demonstrated.
What is small amount of nickel? In which form is nickel added? What is the role of nickel?
Line 94: Which type of synthesis of a non-stoichiometric perovskite (Electrochemical, ,,?
Line 97: "stability for the fine exsolved nanoparticles embedded in the perovskite substrate". Which type of nanoparticles?
Line 125:" fine core-shell particles encapsulated on the surface". What is core, what is shell in structure? What is particle size of core and shell?
Line 132: (calcination at 500 °C and reduction at 800 °C) ? Calcination of nickel nitrate? What is product of calcination reaction? What is size of nickel particles?
Line 136: Please to write the meaning of TPR and TEM
Line 140. The number of publication is missing
Line 149: Temperature (°C) as X-axis is missing at Figure 2. In which unit is hydrogen rate consumption? Which type of EDS analysis was used? What is parameter at X-axis of EDS Analysis? (particle size or thickness or length?)
Line 153: BET analysis (the meaning of BET analysis?)
Line 194: Please to separate H2 and (%) in Table 1
Line 199: electrolytes (e.g. CGO [49], LSGM [54]and BYCO; Can you give a meaning of 3 mentioned electrolytes?
Line 220: What is the meaning of CGO?
Line 227: complete peroviste-structure; correct "perovskite structure"
Line 262: What are internal redox properties of the ceria?
Line 354: "Moreover, a limited coarsening effect is generally observed for the nanoparticles even..."Which nanoparticles?
General comments:
- How is synthetisized lanthanum ferrites-based exsolved perovskites? What are precursors for this synthesis of La0.6Sr0.4Fe0.8Co0.2O3
- How is possible to reach the stoichiometry from precursor to the product during synthesis?
- Which characterisation method was used for the Figure 2 (High resolution transmission electron microscopy?)
- Is shuttle mechanism proposed by authors or from literature?
- In which reaction time is a shuttle mechanism valid? Is there some limitation for this shuttle mechanism?
Author Response
A. We thank the reviewer for the useful suggestions that give us the opportunity to ameliorate the manuscript. Accordingly, we reply point-by-point below to the specific observations.
R. Line 15 "of the nanoparticles exsoluted on the substrate surface can be further modulated by a post treatment". Which nanoparticles?; Which size?
A. This sentence has been deleted according to the reviewer’s comment.
R. Line 18: at the SOFC anode (please to write "solid oxide fuel cell (SOFC)
A. Thank you, this is now corrected
R. Line 26: Which core-shell nanoparticles?
A. This term has been replaced with exsolved nanoparticles
R. Line 86: perovskites commonly used as a SOFC cathode with small amounts of Ni has been demonstrated.
A. The sentence is “Proper modification of commercial type ferrite-based perovskites, commonly used as a SOFC cathode, with small amounts of Ni has been demonstrated”. This refers to the LSFC as one of the currently used cathodes in SOFCs.
R. What is small amount of nickel? In which form is nickel added? What is the role of nickel?
A. The amount of Nickel addition is 3 wt. % (now specified). This is added as Ni nitrate and calcined (lines 114-118). The role of nickel is to promote the exsolution mechanism (lines 118-121). Some papers report on the role of Ni containing oxides playing a catalytic role, as reported at lines 123-126. As reported at lines 202-205, the catalytic behaviour of this exsolved perovskite was different than the behaviour reported in the literature for the bare LSFC. In addition, we have added a sentence at lines 220-221 related to the elextrochemical behaviour of LSFC and Ni-LSFC for the electroxidation of dry propane as previously investigated (Electrochemical investigation of a propane-fed solid oxide fuel cell based on a composite Ni-perovskite anode catalyst. Applied Catalysis B-Environmental 2009, 89, 49-57, doi:10.1016/j.apcatb.2008.11.019).
R. Line 94: Which type of synthesis of a non-stoichiometric perovskite (Electrochemical, ,,?
A. As reported in the literature (In situ exsolution of PdO nanoparticles from non-stoichiometric LaFePd0.05O3+Δ electrode for impedancemetric NO2 sensor. Sensors and Actuators, B: Chemical, Volume 298, 1 November 2019, Article number 126827), the occurrence of a non-stoichiometric composition for the host oxide is a possible strategy for the preparation of a decorated perovskite. We have added this new reference to the revised text.
R. Line 97: "stability for the fine exsolved nanoparticles embedded in the perovskite substrate". Which type of nanoparticles?
A. The exsolution process involves the occurrence of nanoparticles on the surface as observed from TEM analysis. This is now clarified.
R. Line 125:" fine core-shell particles encapsulated on the surface". What is core, what is shell in structure? What is particle size of core and shell?
A. This is clarified by the subsequent sentence: “…These are composed of a tri-metallic alloy (e.g. Ni-Fe-Co) in the core and a shell of mixed oxides (e.g. α-Fe100-y-zCoyNizOx)). Concerning the particle size, this is now specified in the revised text at line 137.
R. Line 132: (calcination at 500 °C and reduction at 800 °C) ? Calcination of nickel nitrate? What is product of calcination reaction? What is size of nickel particles?
A. The description is provided in the caption of fig.1 and described in the text at lines 12-13 (abstract) and lines 116-118. This thermal treatment has been carried out on the impregnated LSFC in order to promote the exsolution of fine particles. The size of the particles, as determined from the TEM analysis, was about 30 nm as reported at the line 137.
R. Line 136: Please to write the meaning of TPR and TEM
A. Thank you for this observation. TEM has been defined at the line 134, TPR at the line 150, and EDX at line 134
R. Line 140. The number of publication is missing
A. Thank you. We have revised the text accordingly.
R. Line 149: Temperature (°C) as X-axis is missing at Figure 2. In which unit is hydrogen rate consumption? Which type of EDS analysis was used? What is parameter at X-axis of EDS Analysis? (particle size or thickness or length?)
A. Thank you for this observation. The X-axis is now defined. The TPR analysis generally provides an intensity signal (e.g. counts per second proportionally to H2 consumption). This is converted from the native software of this instrument on the basis of the voltage variation occurring on the Winston bridge of TCD in counts per second. We prefer reporting this intensity signal in arbitrary units (a.u.). The additional information has been determined by using an HR-TEM. The parameter on the X-axis of EDX is referred to the length of the cross-section of a nanoparticle. EDX analysis refers to the energy dispersive analysis of X-rays emitted by the sample due to the high energy electron bombardment. This analysis has been carried out during high-angle annular dark-field (HAADF) imaging of the sample. This is now specified.
R. Line 153: BET analysis (the meaning of BET analysis?)
A. BET is now defined at the line 172
R. Line 194: Please to separate H2 and (%) in Table 1
A. This is done in the revised text.
R. Line 199: electrolytes (e.g. CGO [49], LSGM [54]and BYCO; Can you give a meaning of 3 mentioned electrolytes?
A. CGO has been defined in the line 151, LSGM and BYCO were defined in the abstract (lines 19-20). In addition, the revised text now includes a glossary of acronyms.
R. Line 220: What is the meaning of CGO?
A. CGO is Ce0.8Gd0.2O2-d. It is now defined in the line 151.
R. Line 227: complete peroviste-structure; correct "perovskite structure"
A. Thank you, this has been corrected
R. Line 262: What are internal redox properties of the ceria?
A. The complete sentence is: “the influence of the internal redox properties of the ceria based electrolyte”. The ceria based electrolyte, i.e. CGO or others, shows a redox mechanism due to the equilibrium between two oxidation states Ce4+ and Ce3+. The most effective electronic state depends on the reducing force. As consequence, the oxidation state on the cathode is Ce4+, whereas on the anode side this is Ce3+. The migration of oxygen from the cathode to the anode has the effect to move the equilibrium between these two oxidized states of Ce. This corresponds to an internal chemical “short circuit” for the CGO electrolyte. This mechanism is quite well known and it is discussed in the references included in the text (lines 290-294).
R. Line 354: "Moreover, a limited coarsening effect is generally observed for the nanoparticles even..."Which nanoparticles?
A. We have now elucidated properly this sentence (line 450)
R. General comments:
How is synthetisized lanthanum ferrites-based exsolved perovskites? What are precursors for this synthesis of La0.6Sr0.4Fe0.8Co0.2O3
A. The LSFC used for the tests reported in this work is a commercial powder. It is reported at lines 109-113 and line 400-401 of the revised text.
R. How is possible to reach the stoichiometry from precursor to the product during synthesis?
A. The amount of Ni added to the perovskite was evaluated on the basis of a compromise between the need of avoiding large occurrence metallic nickel on the surface with the consequent risk of carbon deposition during cracking of organic fuels and favouring just a partial exsolution of Fe and Co from the perovskite. As consequence, the 3 wt. % of Ni used for this electrocatalyst was not related to any stoichiometric evaluation.
R. Which characterisation method was used for the Figure 2 (High resolution transmission electron microscopy?)
A. HR term has been added in the revised text
R. Is shuttle mechanism proposed by authors or from literature?
In which reaction time is a shuttle mechanism valid? Is there some limitation for this shuttle mechanism?
A. Shuttle mechanism is reported in previously reported papers (e.g. Solid oxide fuel cells fed with dry ethanol: The effect of a perovskite protective anodic layer containing dispersed Ni-alloy @ FeOx core-shell nanoparticles. Applied Catalysis B: Environmental 2018, 220, 98-110, doi:https://doi.org/10.1016/j.apcatb.2017.08.010).
This term was originally used by our group to indicate the migration of H2 and CO, produced at the pre-catalytic layer according to an internal reforming step, to the interface and their return to the pre-catalytic layer as H2O and CO2 favouring the fuel processing. This is now clarified. As may be derived from the EIS analysis, the typical reaction frequency for the slowest oxidation step is around a few Hz. Such data has been evaluated for the oxidation of ethanol and reported in the paper: Electrochemical investigation of a propane-fed solid oxide fuel cell based on a composite Ni-perovskite anode catalyst. Applied Catalysis B-Environmental 2009, 89, 49-57, doi:10.1016/j.apcatb.2008.11.019. Nevertheless, since we have not evaluated this in detail for all the fuels here investigated, we prefer to avoid to discuss this aspect. Mass transfer constraints may results from the shuttle mechanism and have a negative impact on polarization curves at high current density.
Reviewer 3 Report
The manuscript entitled “A mini review on lanthanum ferrites-based exsolved perovskites as fuel-flexible anode for Solid Oxide Fuel Cells” is relatively well written with nice Figures. However, the authors should critically discuss the perspectives and future challenges. I propose the addition of a new subsection entitled “Key insights on challenges and perspectives”. The authors have to point out the challenges and future directions.
Author Response
R. The manuscript entitled “A mini review on lanthanum ferrites-based exsolved perovskites as fuel-flexible anode for Solid Oxide Fuel Cells” is relatively well written with nice Figures. However, the authors should critically discuss the perspectives and future challenges. I propose the addition of a new subsection entitled “Key insights on challenges and perspectives”. The authors have to point out the challenges and future directions.
A. Many thanks for this suggestion. We have improved the manuscript by including a further chapter in which we discuss a new approach where an exsolved LSFC perovskite is modified with a NiCu alloy. This approach allowed to address the low reactivity of exsolved perovskite towards methane making the pre-catalytic layer more suitable for a fuel-flexible SOFC. This represents a future direction. Moreover, some challenges of the developed approach in terms of mass transfer issue have been discussed
Reviewer 4 Report
This paper presents a mini review about the preparation methods, physico-chemical characteristics, surface properties and electrochemical properties of lanthanum ferrites-based exsolved perovskites as fuel-flexible anode for Solid Oxide Fuel Cells. Although this paper contributes to some novel knowledge to the related field, it needs some improvements before acceptance for publication. The major comments are as follows:
(1) This manuscript was submitted to Materials to be considered for publication but there are no references to this journal in the bibliography. It is some surprising that the authors decide to send the manuscript to be considered for publication to a journal where they have found no references to cite and to compare (if possible) the reported results with those found in the new references.
(2) Some mistakes in the redaction of the manuscript must be corrected. For example, line 66 (material oxygen storage material), , etc.
(3) American english and British english are both used in the manuscript. Please select one and correct the other expressions to unify the grammar used.
(4) Line 98: as this submission is a mini review, it should be understood by a specialised and a non-specialised audience on SOFC. Sometimes, the review manuscripts are sources of information to start a research line in the issue. Therefore, I recommend to explain (just a little bit) the meaning of "A and B sites".
(5) Line 108: in the wet impregnation method, a 3% of Ni as Ni nitrate was deposited drop-by-drop at 80ºC. In order to describe a reproducible method, please add the concentration of the Ni solution and the volume used until the impregnation was finished.
(6) Some acronyms must be defined. As the manuscript presents a high number of different acronyms, I suggest to include a section of acronyms meaning at the beginning of the article, and to delete some definitions provided in the text (e.g. line, 161: autothermal reaction (ATR)).
(7) Table 2 (and its discussion). Some of the maximum power density, series resistance and total resistance values are quite similar. In this sense, I think that the stadistical deviation of each test must be included in the results in order to achieve a better comparison. Ethanol and Propane are clearly the best SOFC in terms of the maximum duratiblity demonstrated and the average decay during the life time, but I think that a further discussion can be done if the errors of each value are reported.
(8) There are some typo mistakes. E.g. line 325: "oxidation dry" must be "oxidation of dry".
(9) Section no.5: "Reaction Mechanism". This section is very incomplete since there are no chemical reactions to describe a true reaction mechanism. In addition, there are no references to justify the steps proposed. I strongly recommend to work hard in this section to improve it.
(10) References: only 15 of the 90 references are based on the last 5 years publications. Therefore, I recommend to add some more actual references.
Author Response
A. We thank the reviewer for the useful suggestions that have given us the opportunity to ameliorate the manuscript. In the following, we reply point-by-point to the reviewer’s observations.
R. This manuscript was submitted to Materials to be considered for publication but there are no references to this journal in the bibliography. It is some surprising that the authors decide to send the manuscript to be considered for publication to a journal where they have found no references to cite and to compare (if possible) the reported results with those found in the new references.
A. Many thanks for this observation. We have now cited some previous papers from this journal. This is a relatively new topic for Materials. We expect it will attract the wide interest of the readers. The topic is specifically addressing to new materials for SOFCs.
R. Some mistakes in the redaction of the manuscript must be corrected. For example, line 66 (material oxygen storage material), , etc.
A. Thank you. A proper check has been carried out for the manuscript.
R. American english and British english are both used in the manuscript. Please select one and correct the other expressions to unify the grammar used.
A. We have revised the text accordingly.
R. Line 98: as this submission is a mini review, it should be understood by a specialised and a non-specialised audience on SOFC. Sometimes, the review manuscripts are sources of information to start a research line in the issue. Therefore, I recommend to explain (just a little bit) the meaning of "A and B sites".
A. We have added a description accordingly (lines 72-77). Furthermore, the title was modified.
R. Line 108: in the wet impregnation method, a 3% of Ni as Ni nitrate was deposited drop-by-drop at 80ºC. In order to describe a reproducible method, please add the concentration of the Ni solution and the volume used until the impregnation was finished.
A. We have added these details (line 114)
R. Some acronyms must be defined. As the manuscript presents a high number of different acronyms, I suggest to include a section of acronyms meaning at the beginning of the article, and to delete some definitions provided in the text (e.g. line, 161: autothermal reaction (ATR)).
A.Thank you for this observation. We have preferred to define acronyms through the text (also according to the suggestions raised from the other reviewers) and we have provided a list as well.
R. Table 2 (and its discussion). Some of the maximum power density, series resistance and total resistance values are quite similar. In this sense, I think that the statistical deviation of each test must be included in the results in order to achieve a better comparison. Ethanol and Propane are clearly the best SOFC in terms of the maximum durability demonstrated and the average decay during the life time, but I think that a further discussion can be done if the errors of each value are reported.
A. The standard deviation is indicated in the figure captions
R. There are some typo mistakes. E.g. line 325: "oxidation dry" must be "oxidation of dry".
A. The overall text has been revised accordingly
R. Section no.5: "Reaction Mechanism". This section is very incomplete since there are no chemical reactions to describe a true reaction mechanism. In addition, there are no references to justify the steps proposed. I strongly recommend to work hard in this section to improve it.
A. The reaction mechanism was thoroughly discussed in our previous papers. We have provided two specific references for the readers and we discussed the mechanism by including a figure (fig.5) representing what we intend as a “shuttle mechanism”. It is noteworthy that the mechanism suggested in these papers, was supported by the chromatographic analysis of effluents. We have however accepted the reviewer’s suggestion and the text has been revised accordingly (lines 363-378).
R. References: only 15 of the 90 references are based on the last 5 years publications. Therefore, I recommend to add some more actual references.
A. Some recent references have been added, accordingly
Round 2
Reviewer 2 Report
Thank you for your answers. This improved Version can be accepted in present form.
Reviewer 3 Report
Accept in present form
Reviewer 4 Report
The authors have updated the manuscript in accordance with my suggestions. Therefore, I recommend to accept this paper in its actual form.